# A probabilistic knowledge graph for target identification

**Chang Liu**[1☯], **Kaimin Xiao**[2,3☯], **Cuinan Yu**[4☯], **Yipin Lei**[1☯], **Kangbo Lyu**[1], **Tingzhong Tian**[1], **Dan Zhao**[1]*, **Fengfeng Zhou**[5]*, **Haidong Tang**[2]*, **Jianyang Zeng**[6,7,8]*

**1** Institute for Interdisciplinary Information Sciences, Tsinghua University, Beijing, China, **2** School of Pharmaceutical Sciences, Tsinghua University, Beijing, China, **3** Joint Graduate Program of Peking-Tsinghua-NIBS, School of Life Sciences, Tsinghua University, Beijing, China, **4** Machine Learning Department, Silexon AI Technology Co., Ltd., Nanjing, Jiangsu Province, China, **5** Key Laboratory of Symbolic Computation and Knowledge Engineering of Ministry of Education, College of Computer Science and Technology, Jilin University, Changchun, Jilin Province, China, **6** School of Engineering, Westlake University, Hangzhou, China, **7** Westlake Laboratory of Life Sciences and Biomedicine, Hangzhou, China, **8** Research Center for Industries of the Future and School of Engineering, Westlake University, Hangzhou, Zhejiang Province, China

☯ These authors contributed equally to this work.
* zhaodan2018@tsinghua.edu.cn (DZ); FengfengZhou@gmail.com (FZ); hdtang@mail.tsinghua.edu.cn (HT); zengjy@westlake.edu.cn (JZ)

**Data Availability Statement:** The source code and data used in this study can be accessed at https://github.com/Dr-Patient/Progeni.

**Funding:** This work was supported in part by the National Natural Science Foundation of China

## Abstract

Early identification of safe and efficacious disease targets is crucial to alleviating the tremendous cost of drug discovery projects. However, existing experimental methods for identifying new targets are generally labor-intensive and failure-prone. On the other hand, computational approaches, especially machine learning-based frameworks, have shown remarkable application potential in drug discovery. In this work, we propose Progeni, a novel machine learning-based framework for target identification. In addition to fully exploiting the known heterogeneous biological networks from various sources, Progeni integrates literature evidence about the relations between biological entities to construct a probabilistic knowledge graph. Graph neural networks are then employed in Progeni to learn the feature embeddings of biological entities to facilitate the identification of biologically relevant target candidates. A comprehensive evaluation of Progeni demonstrated its superior predictive power over the baseline methods on the target identification task. In addition, our extensive tests showed that Progeni exhibited high robustness to the negative effect of exposure bias, a common phenomenon in recommendation systems, and effectively identified new targets that can be strongly supported by the literature. Moreover, our wet lab experiments successfully validated the biological significance of the top target candidates predicted by Progeni for melanoma and colorectal cancer. All these results suggested that Progeni can identify biologically effective targets and thus provide a powerful and useful tool for advancing the drug discovery process.

(T2125007 to J.Z., 32270640 to D.Z., and 82073161, 32270982, 82241234 to H.T.), the National Key Research and Development Program of China (2021YFF1201300 to J.Z.), the New Cornerstone Science Foundation through the XPLORER PRIZE (J.Z.), the Research Center for Industries of the Future (RCIF) at Westlake University (J.Z.), the Westlake Education Foundation (J.Z.), the "Pioneer" and "Leading Goose" R&D Program of Zhejiang (2024SSYS0036), the National Youth Talent Support Program (to H.T.), the Senior and Junior Technological Innovation Team (20210509055RQ), the Fundamental Research Funds for the Central Universities, JLU and the Jilin Provincial Key Laboratory of Big Data Intelligent Computing (20180622002JC). The funders played no role in the study design, data collection and analysis, decision to publish, or preparation of the manuscript.

**Competing interests:** I have read the journal's policy and the authors of this manuscript have the following competing interests: J.Z. is the founder of Silexon AI Technology Co., Ltd. and has an equity interest.

## Author summary

The selection of a disease target, a biological entity with which potential drugs interact to treat diseases, is often the first step of a drug discovery project. Known to be extremely costly, drug discovery projects spend tremendous resources on failed drug instances, most of which result from inadequate choices of targets. To help address the costliness problem of drug discovery, we developed a machine learning-based framework that identifies biologically effective targets using a probabilistic knowledge graph built from both biological network data and literature evidence. Our method not only outperforms state-of-the-art baseline methods on the target prediction task, but can also identify targets with high biological relevance, as shown by the strong support of the literature for the predicted target candidates and wet lab experiments that validate the significance of the predicted target candidates for melanoma and colorectal cancer. These results suggest that our method can identify effective targets and therefore benefit drug discovery.

## Introduction

A drug discovery project often initiates by developing hypotheses that the activation or inhibition of specific biological entities (e.g., proteins, pathways, and RNA) induces a therapeutic effect in treating diseases [1]. These entities of interest are commonly called "targets." Early identification of safe and effective targets is generally crucial to reducing the massive cost and minimizing the high attrition rates in drug discovery [2–4].

Several experimental methods, such as genetic manipulation (e.g., using RNAi or small-molecule perturbation) [5], genome-wide association studies [6, 7], and transcriptome analyses [8], have made significant advancements in identifying targets. However, these experimental methods are often labor-intensive and prone to failure. On the other hand, large-scale databases, such as DrugBank [9], DisGeNET [10], and the Comparative Toxicogenomics Database [11], contain various biological networks that can be represented as knowledge graphs. These graphs document numerous relations (e.g., associations, interactions, and similarities) between biological entities (e.g., drugs, targets, diseases, and side effects). Such a wealth of graph data provides fertile ground for developing network-based computational methods. These methods learn feature representations of biological entities to predict new interactions or associations (e.g., target-disease associations) even without prior experimental evidence. These network-based methods have demonstrated significant potential in substantially reducing the cost and labor associated with the traditional target identification process [12–14].

Diffusion-based methods, a subset of network-based methods, have been extensively applied to target identification. They involve performing random walks on biological networks associated with targets and diseases, such as protein-protein interaction networks and protein-disease association networks. Subsequently, the stationary distributions obtained from the random walks are used to prioritize the corresponding target genes for specific diseases [12, 15]. These diffusion-based methods achieved state-of-the-art performance for target identification [16] and accurately identified targets linked to Mendelian disorders [13]. Furthermore, diffusion-based models can be easily adapted to incorporate biological networks that describe the diverse relations between drugs, targets, diseases, and side effects. As exemplified by DTINet [17] (originally designed for predicting target-drug interactions), such expansion of input information significantly enhances prediction accuracy. DTINet has also been successfully repurposed for target identification, achieving state-of-the-art results [14].

Nevertheless, the feature learning process in these diffusion-based methods is often disconnected from the target prediction, and thus may not yield the optimal results. Moreover, the stationary distributions that represent the features of biological entities are generally fed to relatively simple learning models (e.g., logistic regression). These models may lack the capacity to capture the complex latent features present in the heterogeneous data [18]. On the other hand, graph neural networks (GNNs) are nonlinear machine learning models that extend convolutional neural networks to process graph data [19, 20]. GNNs incorporate novel techniques for information propagation and aggregation [21, 22], and have demonstrated exceptional performance when applied to tasks involving heterogeneous networks [23–25]. Moreover, GNNs have emerged as an important paradigm for advancing the drug discovery process [26–28]. For instance, NeoDTI [26] achieves state-of-the-art performance in predicting target-drug interactions by using GNNs to leverage heterogeneous biological networks. However, despite the success stories of these computational methods in target identification, they still have the following limitations:

First, the input biological networks typically represent relations as binary, which means that they do not convey the likelihood of these relations being biologically relevant. Hence, the predictions made by these models may not always hold biological significance in the real world. This necessitates extensive manual labor in the experimental validation of the predictions [29]. Second, model performance is often heavily influenced by *exposure bias*, a common phenomenon in recommendation systems. Exposure bias occurs when the unobserved interactions between users and items do not always represent the users' rejection of those items [30]. In such situations, models generally tend to predict fewer relations between entities with limited information (e.g., targets associated with fewer observed diseases). This further induces the challenges in learning the latent feature representations of these entities and consequently weakens the overall prediction performance. Such limitations inevitably undermine the application potential of these prediction models in target identification, thereby impeding their contributions to drug discovery.

To overcome these limitations, we develop Progeni (**PR**obabilistic kn**O**wledge **G**raph for targ**E**t ide**N**tif**I**cation), a novel machine learning-based framework for target identification. In addition to fully exploiting the existing heterogeneous biological networks from various sources, Progeni innovatively integrates literature evidence, which contains vast information about the relations between biological entities, to construct a probabilistic knowledge graph (prob-KG). Graph neural networks (GNNs) are then leveraged to learn the latent feature representations of biological entities in the prob-KG, thus informing the predictions of biologically meaningful targets. A comprehensive evaluation demonstrated that Progeni achieved state-of-the-art performance on the target identification task and showed remarkable robustness against the effect of exposure bias. In addition, Progeni successfully identified new targets with strong supporting evidence from the literature. Moreover, our wet lab experiments successfully validated the biological significance and therapeutic potential of the top novel target candidates predicted by Progeni. All these results demonstrated that Progeni can provide a powerful and useful tool for target identification and thus benefit the drug discovery process.

## Results

### Progeni infers new target candidates based on a probabilistic knowledge graph

Exploiting available information on known targets (hereafter, we will use the terms "target" and "protein" interchangeably) is essential for inferring new target candidates. Biological networks, which document relations between different biological entities (e.g., the interactions or

associations between targets, diseases, drugs, and their side effects), facilitate graph learning methods in capturing latent features of entities to infer new relations [17, 26]. Additionally, unstructured data from literature evidence can also provide valuable information about known relations. For instance, the frequency of literature mentioning a relation, approximated by the frequency of literature mentioning both related entities, indicates the strength of evidence supporting its biological relevance.

In this context, Progeni (Fig 1A) integrates both biological networks and literature evidence to construct a probabilistic knowledge graph (prob-KG). Each node in the graph represents a biological entity (e.g., target, disease, drug, and side effect), and each edge stands for a relation (e.g., association or interaction) between two entities observed in the biological networks. Moreover, each edge is assigned a probability score derived according to the co-occurrence frequency of the two entities in the literature evidence (Fig 1B, Methods, and S1 Text). Target identification is then formulated as a missing link prediction task on the defined prob-KG.

Progeni employs a graph neural network (GNN) approach to infer new target candidates based on the constructed prob-KG (Fig 1C). First, Progeni aggregates the information (i.e., initial feature embeddings) of neighboring nodes via a graph convolution step, deploying different GNNs for individual types of relations (i.e., edges). For example, three separate GNNs are

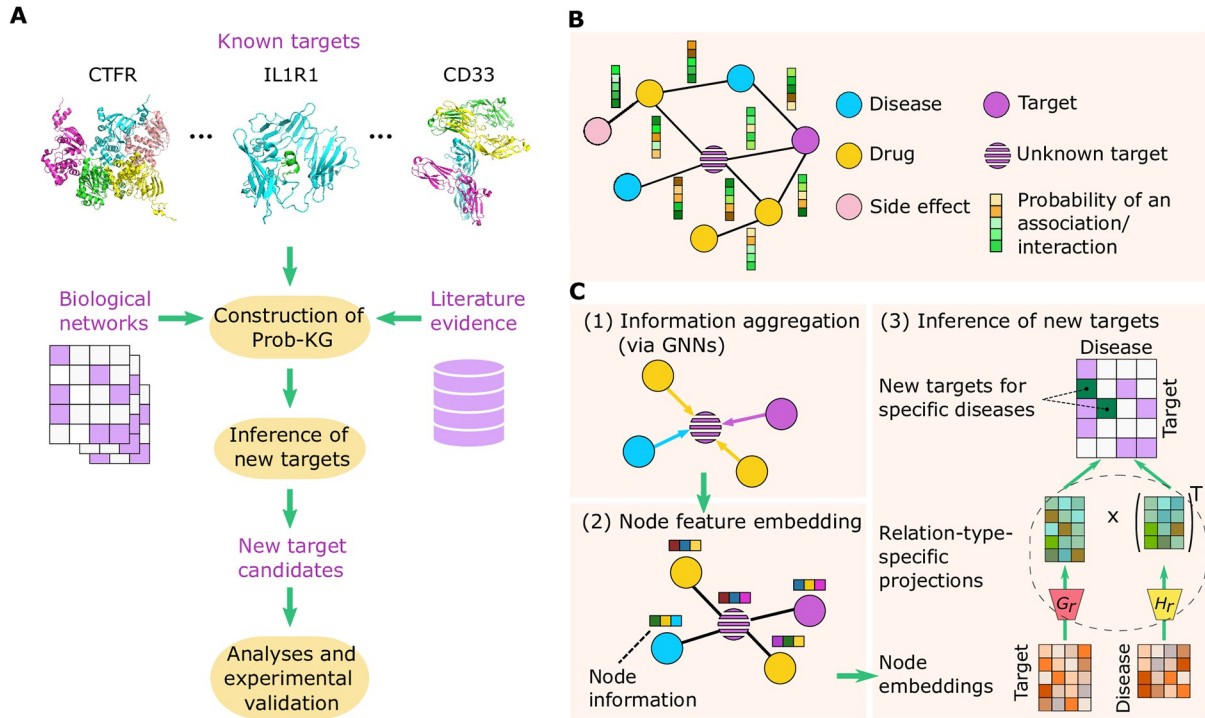

**Fig 1. Overview of the Progeni pipeline.** (**A**), Information flow of Progeni. Progeni first integrates a list of known targets, various biological networks, and literature evidence to construct a probabilistic knowledge graph (prob-KG), which is then used to infer new target candidates. The new target candidates inferred by Progeni are finally analyzed and experimentally validated. (**B**), Construction of the probabilistic knowledge graph (prob-KG). The structure of the prob-KG is obtained through integrating various biological networks, which document the associations/interactions between different biological entities, e.g., diseases, drugs, targets, and side effects. Each edge represents an association/ interaction between two entities and is associated with a probability score assigned according to the co-occurrence frequency between the two entities derived from literature evidence. (**C**), Inference of new targets. The information of neighboring nodes is first aggregated via graph neural networks (GNNs) employed for individual relation types. Next, the aggregated node information is projected onto a feature vector space to generate the embedded node features. Finally, the learned node embeddings are used to reconstruct the prob-KG (as the optimization objective) via relation-type-specific projections, whereby the new target candidates are inferred. More details about the Progeni framework can be found in Methods.

used for a drug node to aggregate information from neighboring drug, disease, and target nodes, respectively. After summing up the aggregated relation-type-specific neighborhood information, Progeni then projects the resulting node information onto a feature vector space to obtain the embedded node features.

Next, Progeni maps the node feature embeddings to edge probability matrices via relation-type-specific projections (Fig 1C, Methods). Specifically, through a pair of transformation matrices, each projection operation maps the node feature space to an edge probability space. In its optimization objective, Progeni minimizes a weighted sum of the differences between the mapped and ground truth edge probabilities. Here, since the edges with higher probabilities generally represent the relations with more biological relevance, Progeni assigns larger weights to the corresponding differences to encourage the model to assign higher confidence to these relations. Conversely, since the edges with lower probabilities generally represent the relations with higher uncertainty, Progeni assigns smaller weights to the corresponding differences to avoid overfitting to the potentially noisy data.

After training, the reconstructed edge probabilities of the target-disease association (TDA) network are retrieved and organized into a matrix. The reconstructed edge probability of an element in the TDA matrix serves as the prediction score of a potential target for the corresponding associated disease. More details about the Progeni pipeline can be found in Methods.

## Progeni accurately identifies disease targets

We evaluated Progeni through cross-validation tests, formulated as a missing link prediction task for the target-disease association (TDA) relations, in which the goal is to predict each target-disease pair as associated or non-associated. A heterogeneous biological network dataset [17] was used to evaluate the performance of our model and several baseline methods that had reached state-of-the-art performance on the missing link prediction tasks on heterogeneous graphs, including DTINet [17], GTN [24], RGCN [23], and HGT [25] (see Methods and S1 Text for more details about the data and these baselines). The reconstructed edge probability matrix of TDA relations was used as the found predictions. The ground truth label of one reconstructed edge was one if the corresponding TDA was observed in the biological networks (i.e., a positive association) and zero otherwise (i.e., a negative association). We measured the performance of the models in terms of the area under the receiver operating characteristic curve (AUROC) and the area under the precision-recall curve (AUPR).

In our performance evaluation, we mainly used two five-fold cross-validation schemes, i.e., entry-wise cross-validation and cluster-wise cross-validation (Fig 2A, Methods). In the entry-wise cross-validation, we randomly partitioned the dataset with respect to individual entries of the edge probability matrix. Our comparison tests showed that Progeni yielded superior performance over the baseline models in the entry-wise cross-validation setting (Fig 2B). However, with Progeni achieving near-perfect performance, this prediction task may seem trivial. To mimic a more realistic application scenario, we further evaluated Progeni and the baselines in a new setting (i.e., cluster-wise cross-validation), where we first clustered the columns of the edge probability matrix according to their similarities and then partitioned the dataset with respect to the resulting clusters (Methods). This clustering operation reduced the similarity between training and test data and thus provided a more realistic and challenging scenario for evaluating different target identification approaches. We found that all models yielded significantly poorer performance when switching from entry-wise cross-validation to its cluster-wise counterpart, attesting to the greater difficulty of the latter. Nevertheless, as shown in Fig 2C, Progeni still took the lead in performance (though DTINet yielded a comparable AUROC score with Progeni, the former achieved a lower AUPR score).

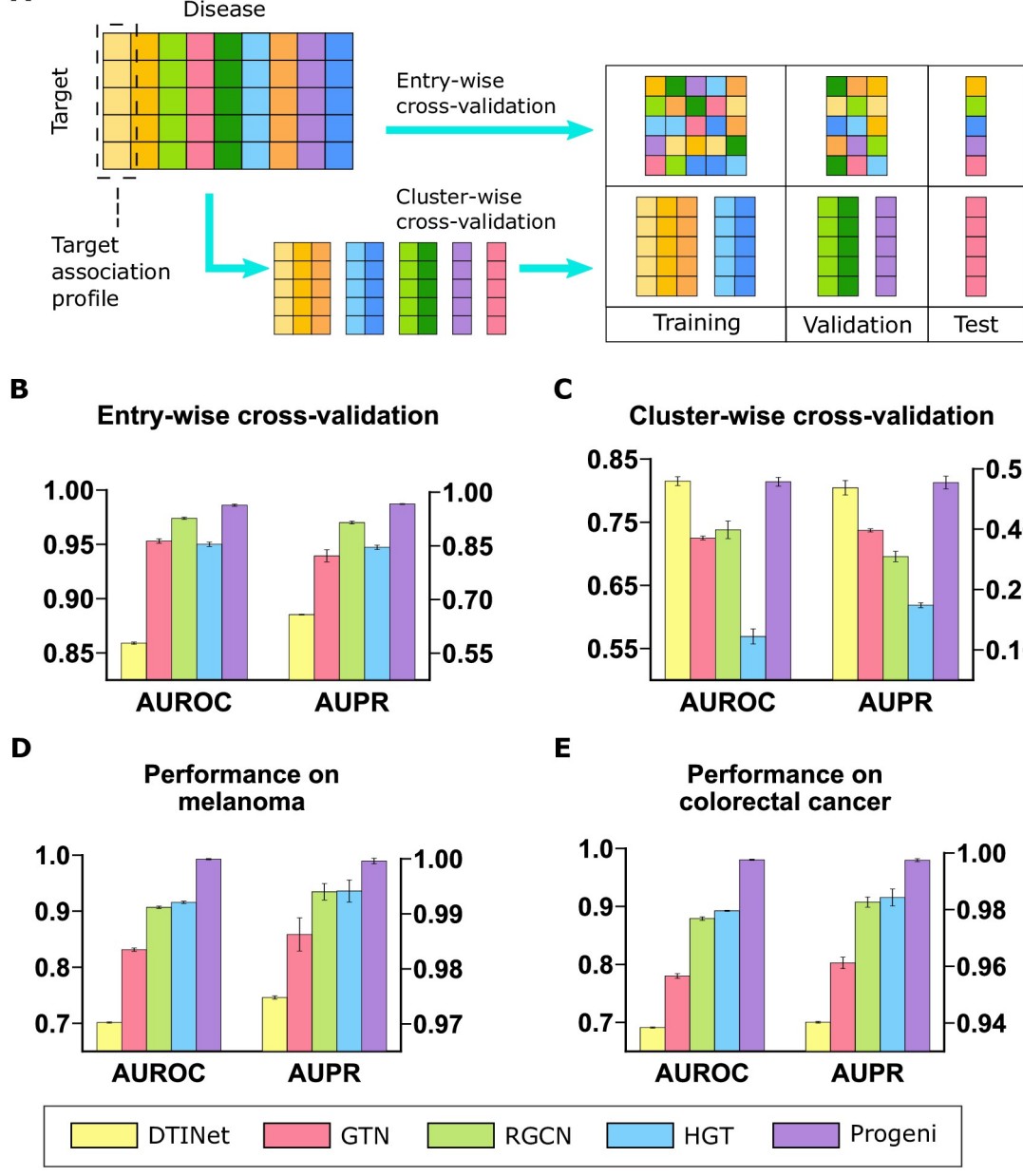

**Fig 2. Performance evaluation on the target-disease association prediction task.** **(A)**, Illustration of the two cross-validation schemes used (see main text for more details). Columns with similar colors belong to the same cluster. **(B)-(C)**, Performance of different models on the entry-wise cross-validation test **(B)** and the cluster-wise cross-validation test **(C)**, respectively. **(D)-(E)**, Performance of of different models trained on the whole prob-KG on target identification for melanoma **(D)** and colorectal cancer **(E)**, respectively. The models used the optimal hyperparameters derived from the cluster-wise cross-validation test setting. All results were summarized over ten trials and expressed as mean ± SD.

To further demonstrate the superior predictive power of Progeni, we also re-trained all models on the whole prob-KG using the optimal hyperparameters derived from the cluster-wise cross-validation test setting and then measured their performance on the target association profiles of two specific diseases, including melanoma and colorectal cancer (CRC). We found that Progeni again identified the targets for these two diseases more accurately than the baseline models (Fig 2D and 2E).

In addition, in our ablation studies, we developed four control models (see Methods for more details) that negated the literature evidence integrated into the edge probabilities and/or the weights in our optimization objective, including Progeni_og (i.e., no literature evidence integrated), Progeni_rp (i.e., using random edge probabilities), Progeni_rw (i.e., using random loss weights), and Progeni_rpw (i.e., using both random edge probabilities and random loss weights). Compared to these control models, Progeni achieved superior performance in the cluster-wise cross-validation setting (Table C in S1 Text), thus well validating the significance of the literature evidence in our framework for target identification.

## Progeni exhibits robustness against the effect of exposure bias

Exposure bias is a common phenomenon in recommendation systems, in which the unobserved interactions are often misrepresented as the users' rejection of the corresponding items [30]. Due to exposure bias, for those entities with few observed interactions or associations, machine learning models may have difficulty in learning their latent feature representations and thus predict fewer meaningful relations between them. This phenomenon also arises in our target identification scenario: for a given disease, those targets with no or few associations observed in the biological networks (and thus in the input prob-KG) do not necessarily lack biological functionality. Exposure bias may yield poor performance for those diseases/targets with few observed TDAs. It may also induce the models to predict a limited number of associated diseases for the targets with little information in the known biological networks. Such an effect may cause the models to leave out biologically significant targets.

Here, we systematically examined the robustness of different models against the effect of exposure bias. We first defined the $k_t$ value for a target as the number of its associated diseases observed in the biological networks, and similarly defined the $k_d$ value for a disease as the number of its associated targets observed in the knowledge graph. We then evaluated the prediction performance on the targets with low $k_t$ values and the diseases with low $k_d$ values for different models trained on the whole prob-KG. Observing that the 10th percentile of $k_t$ values was 567.2 while the medium of $k_d$ values was 11.0, we evaluated the performance (AUPR) of the models on the targets with $k_t < 100$, 300, and 500 and on the diseases with $k_d < 10$, 30, and 50, respectively. We found that in individual scenarios, Progeni yielded better performance compared to the baseline models (Fig 3A and 3B).

Next, we evaluated the Spearman correlations between the target-wise maximum edge probabilities (of the predicted TDA relations) reconstructed by different models and their corresponding $k_t$ values. We found that Progeni yielded a significantly lower correlation than GTN, RGCN, and HGT and a comparable result with that of DTINet (Fig 3C). Here, the correlations indicated how the maximum output values for a target depended on the amount of available information (quantified by the $k_t$ values). Thus, a high correlation generally implied a possible drawback, i.e., the top predictions of the model were likely to leave out the targets that had few observed associations but were nevertheless biologically meaningful. Hence, a weak correlation was generally desirable in this scenario. All the above results showed that Progeni was more robust to the effect of exposure bias, and thus was more likely to discover valuable novel target candidates even under an under-explored setting.

## Progeni identifies biologically meaningful target candidates strongly supported by the literature

We next assessed the strength of literature evidence supporting the target candidates identified by Progeni and the baseline models trained on the whole dataset. Here, the target candidates we analyzed were unobserved in the input biological networks, and thus there were no edges

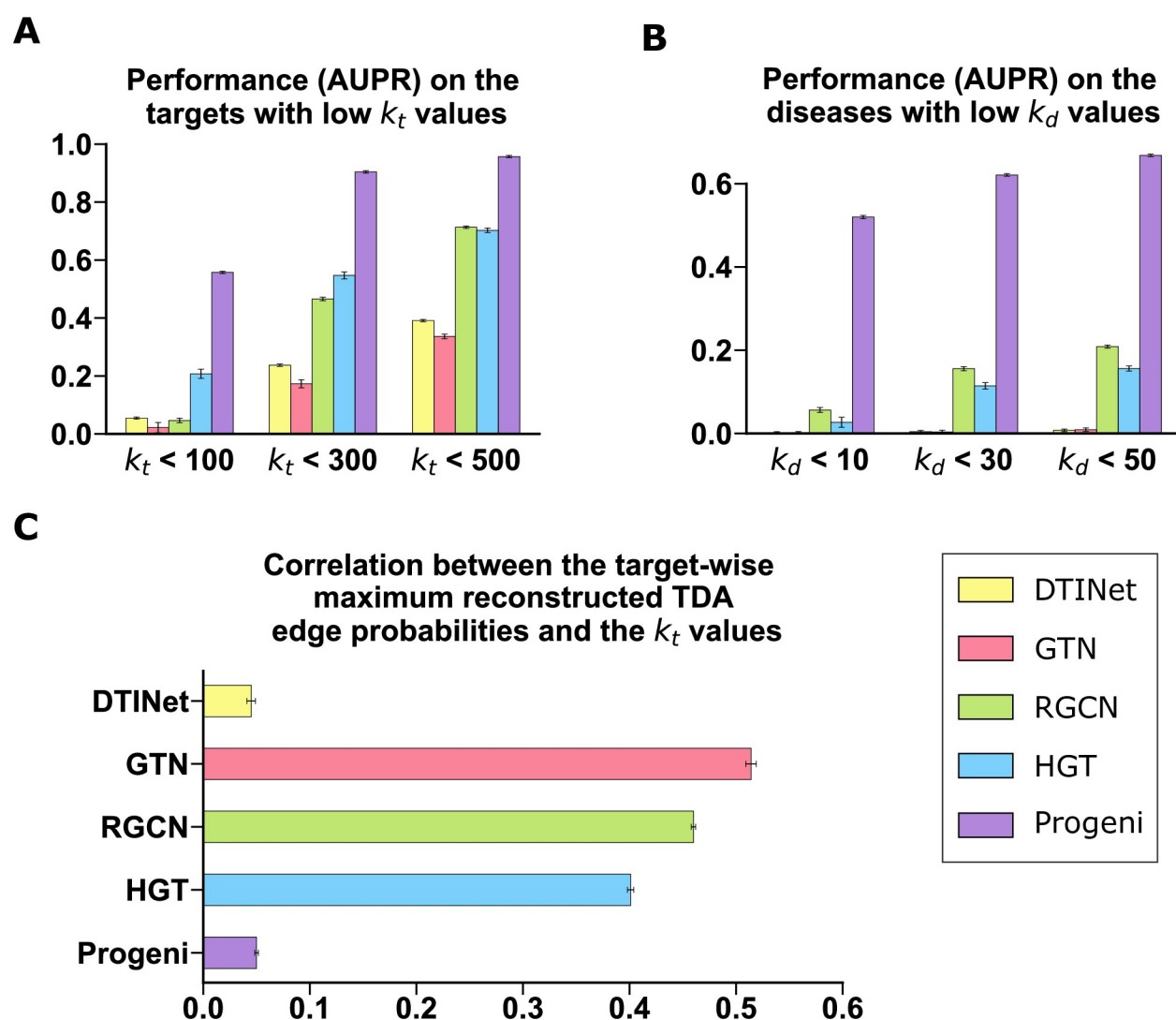

**Fig 3. Evaluation on the robustness of Progeni and baseline models against the effect of exposure bias. (A)-(B)**, The performance of different models on the targets with low $k_t$ (the number of observed associated diseases) values (**A**) and the diseases with low $k_d$ (the number of observed associated targets) values (**B**), respectively. (**C**), The Spearman correlations between the target-wise maximum edge probabilities reconstructed by different models and the corresponding $k_t$ values of targets. All results were summarized over ten trials and expressed as mean ± SD.

between the corresponding disease and target entities in the original input prob-KG. Note that the edge probabilities between these entities were set as zero in the original input prob-KG, thus effectively preventing data leakage during our model training process (see Methods for more details). Nevertheless, there may still exist literature evidence supporting such target-disease associations (TDAs). Therefore, we took advantage of such evidence to evaluate the target candidates (in the form of TDAs) predicted by different models, and those predictions with stronger evidence support can be considered more biologically relevant.

Here, we viewed the reconstructed edge probability between a target and a disease as the prediction score of their association. In particular, we selected a list of predicted TDAs for evaluation according to the following procedure: (i) training the models on the whole prob-KG using the optimal hyperparameters derived from the cluster-wise cross-validation setting; (ii) retrieving the reconstructed edge probability matrix of the TDA relations; (iii) selecting those

*significant* entries in the matrix with respect to each row, in which their prediction scores were greater than $\mu + 2\sigma$, where $\mu$ and $\sigma$ stand for the mean and standard deviation of the scores in that row, respectively; and (iv) choosing the entries representing the TDAs unobserved in the input biological networks.

In our evaluation, the strength of literature evidence supporting the predicted TDA between a target and a disease was approximated by their co-occurrence frequency in the literature, which was denoted by the $C_r$ value (Methods). We first examined the numbers of predicted TDAs with high $C_r$ values. More specifically, for Progeni, the baseline models, and the control models, we first calculated the numbers of predictions with $C_r$ values greater than 0, 5, and 25, respectively, among the list of TDAs with the top-200 prediction scores. We found that in every case, Progeni produced the the highest number of predictions (Fig 4A and 4B). We then investigated the Spearman correlations between the top-$k$ ($k$ = 200, 500, 1500, 2000, 2500, and 3000) prediction scores and their corresponding $C_r$ values. We found that Progeni yielded the highest correlation in every case (Fig 4C and 4D). All these results indicated that compared with the baseline and control models, the target candidates predicted by Progeni received stronger support from the literature and hence were more likely to be biologically meaningful.

To further demonstrate that Progeni can predict reasonable target candidates, we chose the unobserved target candidates with the highest prediction scores for several typical diseases and then verified these target-disease pairs by searching for literature evidence. In particular, we

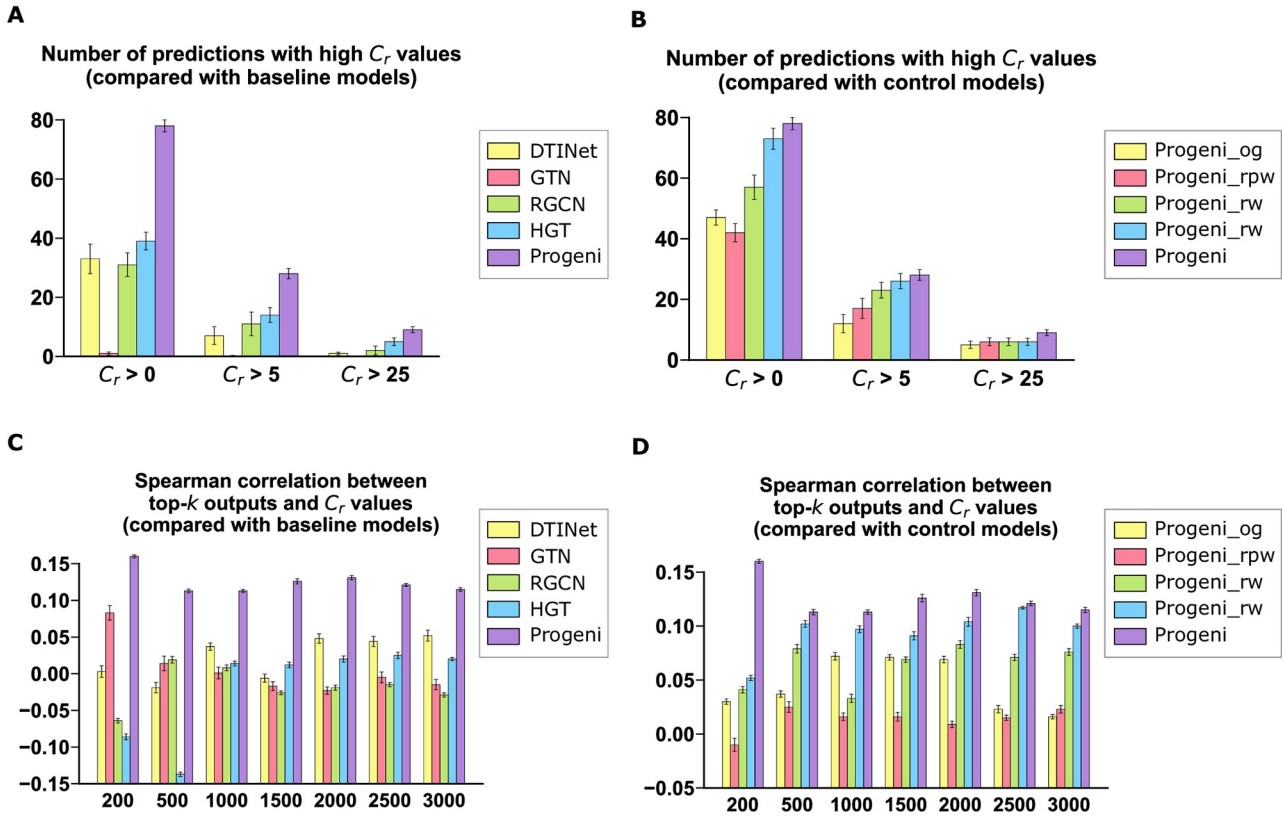

**Fig 4. Evaluation on the strength of literature evidence supporting the target candidates identified by different methods. (A)-(B)**, The numbers of target-disease associations among the top-200 predictions with $C_r$ values, i.e., co-occurrence frequencies, greater than 0, 5, and 25, respectively, for the comparisons of Progeni with different baselines **(A)** and the four control models **(B)**, respectively. **(C)-(D)**, The Spearman correlations between the top-$k$ ($k$ = 200, 500, 1000, 1500, 2000, 2500, and 3000) prediction scores and their corresponding $C_r$ values, for the comparisons of Progeni with different baselines **(C)** and the four control models **(D)**, respectively. All results were summarized over ten trials and expressed as mean ± SD.

presented several case studies for the following diseases: bronchitis, pulmonary arterial hypertension (PAH), and liver neoplasms (Tables D-F in S1 Text). For bronchitis, Progeni identified the cystic fibrosis transmembrane conductance regulator (CFTR) with the third-highest prediction score. It had been previously shown that the bronchitis severity is significantly associated with CFTR activity in the nasal airway [31], lungs [32], and even extrapulmonary tissues [33]. For PAH, the interleukin-1 receptor type 1 (IL1R1) was associated with the highest prediction score among the predictions from Progeni. A previous study had found that *IL1R1* is overexpressed in the lungs of PAH patients, and the hypoxic PAH severity is attenuated among mice treated with an IL1R1 antagonist [34]. Furthermore, Progeni predicted the sialic acid binding Ig-like lectin 3 (SIGLEC-3, or CD33) protein with the second-highest prediction score for liver neoplasms. It had been demonstrated that a *SIGLEC-3* SNP is significantly associated with an increased risk of hepatocellular carcinoma (HCC) among the chronic hepatitis B patients, and the hepatitis B virus activates SIGLEC-3 to induce immunosuppression, thus indicating that the blockade of SIGLEC-3 may provide an effective therapeutic strategy to lower the risk of HCC in the patients with chronic hepatitis B infection [35]. All these findings strongly supported the ability of Progeni in predicting biologically relevant targets, indicating that the therapeutics directed at these targets may constitute efficacious treatments against the associated diseases.

## Progeni predicts novel target candidates validated by wet lab experiments

To further demonstrate the application potential of Progeni, we compiled a new dataset from more up-to-date open sources that contained larger numbers of proteins and diseases (Methods and S1 Text). We trained Progeni on this new dataset and experimentally validated the novel target candidates predicted by Progeni for two specific diseases, i.e., human melanoma and colorectal cancer (CRC). For each disease, we first searched for literature evidence about the top 15 predicted targets unobserved in the known biological networks (Tables G and H in S1 Text) and then chose six target candidates that lacked existing literature support for further experimental validation, including HSP90AB1, RPS6KB1, and MME for melanoma, and ADCY5, ADRA2A, and EEF2 for CRC.

In our *in-vitro* experimental validation, we selected B16F10 (murine melanoma) and MC38 (murine colon adenocarcinoma) as representative cell lines for melanoma and CRC, respectively. We first investigated whether the predicted targets affected cell proliferation by performing CCK-8 assays following the target knockdown by short-hairpin RNAs (shRNAs) in the corresponding cells. The results showed that, in B16F10 cells, the *HSP90AB1* and *MME* knockdowns significantly inhibited cell proliferation, while the *RPS6KB1* knockdown yielded a relatively moderate effect (Fig 5A). In MC38 cells, all the *ADCY5, ADRA2A*, and *EEF2* knockdowns resulted in significant proliferation inhibition (Fig 6A). The raw Optical Density (OD) values for these CCK-8 assays can be found in Tables I and J in S1 Text. These experimental results suggested that the deficiency in these genes predicted by Progeni indeed obstructs tumor cell growth for the corresponding diseases.

Next, we explored the potential correlations between the target gene expression and the clinical outcome of patients from The Cancer Genome Atlas (TCGA). For melanoma, the patients with low *HSP90AB1* and *MME* expression in metastatic tumor samples and the patients with low *RPS6KB1* expression in primary tumor samples were correlated with significantly longer survival than the corresponding high expression cohorts (Fig 5B–5D). Interestingly, the overexpression of *HSP90AB1* [36, 37] and *MME* [38] had been reported to facilitate metastasis, while *RPS6KB1* had been known to play a more crucial role in exacerbating primary tumor [39]. Thus, it is reasonable that the low expression of *HSP90AB1* and *MME* in

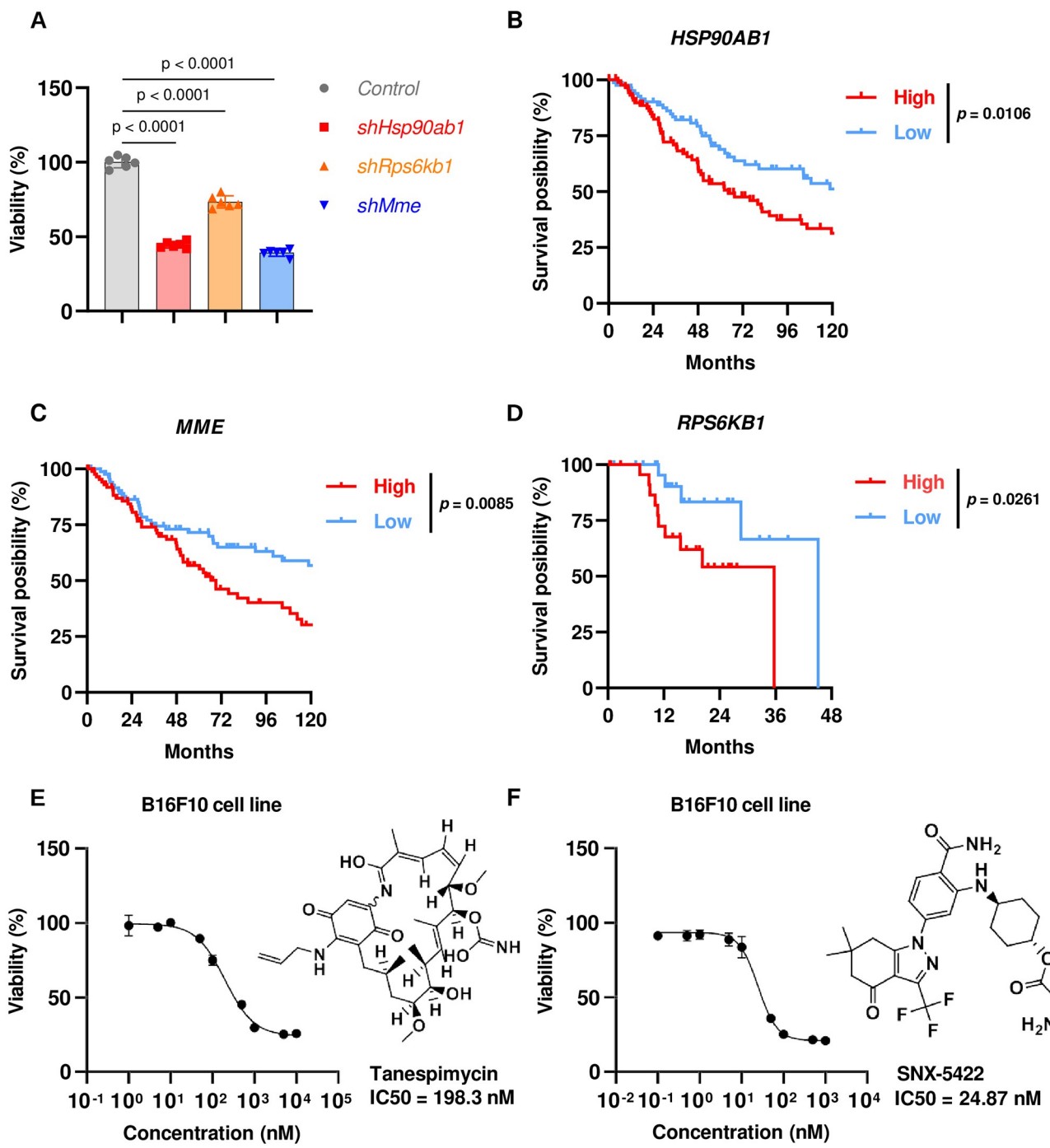

**Fig 5.** *In-vitro* **validation of the target candidates predicted by Progeni for melanoma. (A)**, CCK-8 assays of the B16F10 cells with shRNA knockdown after 36h culture. **(B)-(D)**, Survival curves of the metastatic or primary melanoma patients from The Cancer Genome Atlas (TCGA) with high or low expression of genes *HSP90AB1* (n = 176, metastatic melanoma, **(B)**), *MME* (n = 173, metastatic melanoma, **(C)**), and *RPS6KB1* (n = 50, primary melanoma, **(D)**), respectively. The patients with the 25% highest gene expression were defined as "High," while the lowest 25% were defined as "Low." **(E)-(F)**, CCK-8 assays of the B16F10 cells after 48h treatment with the HSP90AB1 inhibitors tanespimycin **(E)** or SNX-5422 **(F)**, whose molecular structures and half maximal inhibitory concentration (IC50) values are also shown.

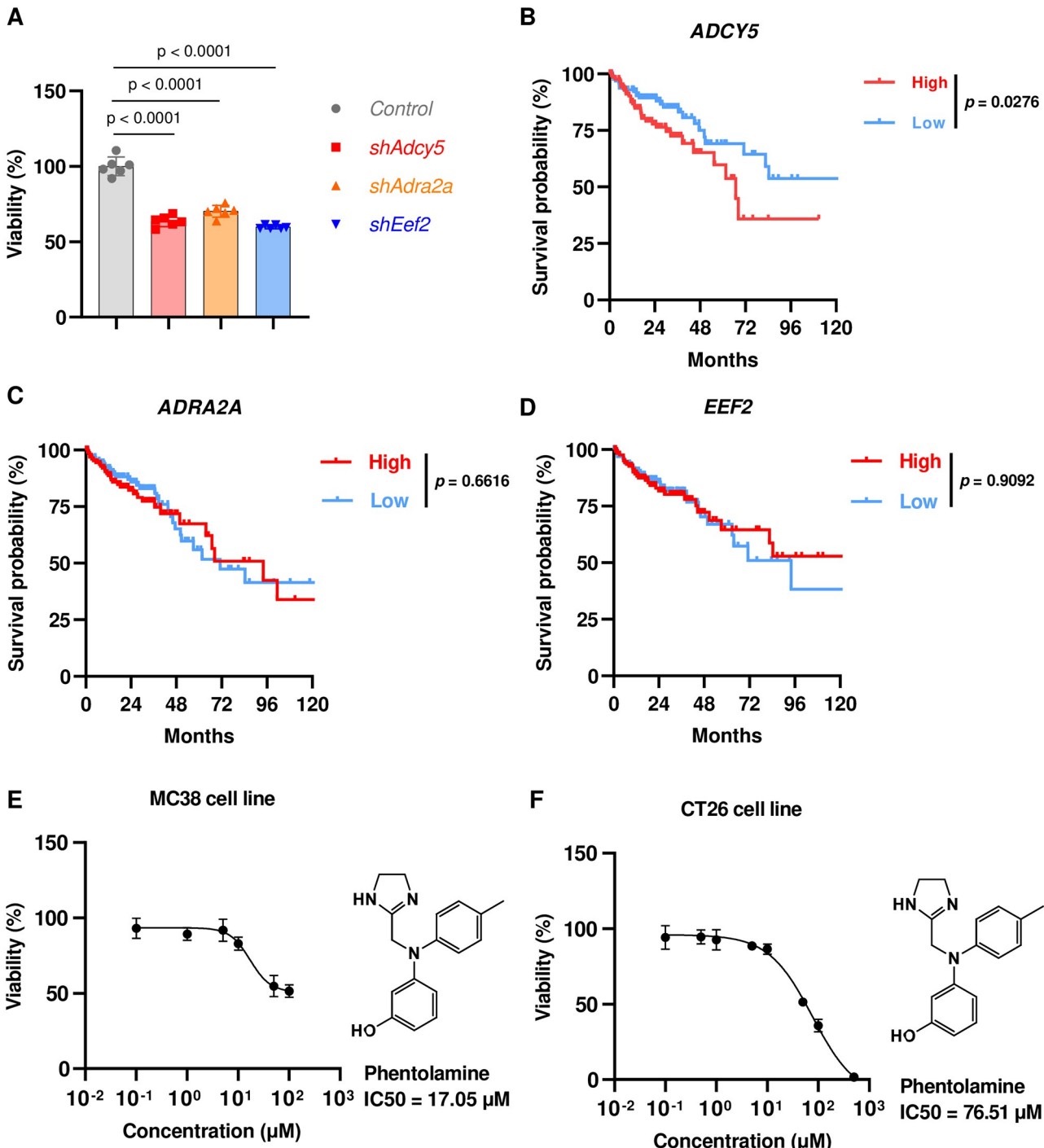

**Fig 6. *In-vitro* validation of the target candidates predicted by Progeni for colorectal cancer (CRC). (A)**, CCK-8 assays of the MC38 cells with shRNA knockdown after 36h culture. **(B)-(D)**, Survival curves of the primary colorectal cancer patients from The Cancer Genome Atlas (TCGA) with high or low expression of *ADCY5* (n = 264, **(B)**), *ADRA2A* (n = 264, **(C)**), and *EEF2* (n = 264, **(D)**), respectively. The patients with the 30% highest gene expression were defined as "High," while the lowest 30% were defined as "Low." **(E)-(F)**, CCK-8 assays of the MC38 **(C)** and CT26 **(D)** cells after 48h treatment with the ADRA2A inhibitor phentolamine, whose molecular structures and half maximal inhibitory concentration (IC50) values are also shown.

metastatic but not primary tumor samples and the low *RPS6KB1* expression in primary but not metastatic tumor samples are correlated with prolonged survival. For the primary colorectal cancer, the low expression of *ADCY5* was significantly associated with longer survival (Fig 6B), whereas no statistically significant differences were found for the corresponding *ADRA2A* and *EEF2* cohorts (Fig 6C and 6D), which may be due to the limited numbers of patients recorded in TCGA (n = 264) that were not sufficient to distinguish the two cohorts of the complex CRC disease. All these results suggested that our predictions were generally consistent with clinical data, while the small sizes of samples from TCGA probably weakened the difference between high and low-expression cohorts.

Furthermore, to evaluate the therapeutic potential of these targets, we also measured the cytotoxicity potency of the small molecule inhibitors targeting them. Tanespimycin and SNX-5422, which are both selective inhibitors of HSP90AB1, significantly inhibited the growth of B16F10 cells after 48 hours of treatment, with a half maximal inhibitory concentration (IC50) of 198.3 nM and 24.87 nM, respectively (Fig 5E and 5F). Phentolamine, an inhibitor of ADRA2A, significantly inhibited the viability of MC38 cells with an IC50 of 17.05 μM and restrained the growth of CT26 cells (another widely used CRC murine cell line) with an IC50 of 76.51 μM (Fig 6E and 6F). The raw Optical Density (OD) values for these CCK-8 assays can be found in Tables K-N in S1 Text. All these results suggested that the novel target candidates identified by Progeni may have great therapeutic potential against the corresponding cancer diseases.

## Discussion

The selection of a disease target is often the first and pivotal step in drug discovery, given the exorbitant cost and high attrition rates that plague the field. In this paper, we have presented Progeni, a novel machine learning-based framework for target identification. Progeni innovatively integrates heterogeneous biological networks and literature evidence to construct a probabilistic knowledge graph, and then leverages graph neural networks to learn the latent feature representations of biological entities.

In our cross-validation tests, while Progeni achieved near-perfect performance when we randomly partitioned the dataset, we observed more reasonable performance when we split the dataset by its clusters. As reflected by the near-perfect result in the former setting, an entry-wise random partition may result in many similar data points in both training and test sets. In the latter setting, data points belonging to the same cluster did not fall into the same fold for cross-validation, which can mimic a more realistic application setting. Our comprehensive evaluation of Progeni demonstrated its remarkable performance in target identification compared to state-of-the-art baseline methods. This superior predictive power will be critical for reducing the cost and increasing the efficiency of drug discovery projects.

A common phenomenon in recommendation systems, exposure bias can impede the prediction of targets with limited information. We developed two tests to gauge the robustness of models against the effect of exposure bias: (i) checking the model performance on those subsets of data with limited information, and (ii) evaluating the correlation between the predicted scores and the amount of information available for individual data points. They are straightforward to implement and may also serve as useful references for future research investigating exposure bias in machine learning-based models. Although we did not develop methods that directly reinforce the model's robustness against the effect of exposure bias, Progeni exhibited remarkable resilience to the adverse effects of exposure bias. This feature indicated that Progeni remains effective, even for targets with limited information in the known biological networks. As a result, Progeni may be better positioned to uncover valuable novel target candidates in those under-explored settings and thus offer a unique advantage in the field.

One of the outstanding qualities of Progeni is its ability to predict biologically meaningful target candidates strongly supported by either literature evidence or wet lab experiments. Compared to baseline and control models, Progeni predicted more target candidates with substantial support from the literature. In our case studies, the literature with co-occurrences between selected diseases and their predicted targets can provide a useful indication about the biological significance of those targets. That said, the current literature validation process still requires manual searching. Future research may invite techniques from natural language processing to automate this process of validating biological significance in the literature. We also compiled a new dataset from more up-to-date open sources to retrain our model and predict novel targets. The fact that our wet lab experiments can successfully validate the predicted targets well reflects Progeni's generalizability to new data. One limitation is that while the experiments used mouse cell lines, our Progeni pipeline only considered human proteins (e.g., when determining the frequency of disease-protein co-occurrence and protein-protein similarity). Another limitation is that we need to fully retrain Progeni for every new dataset, which generally requires additional computational cost. It will be meaningful to reduce such computational cost by eliminating the need for a full retraining process in future research.

In summary, our research has demonstrated that Progeni excels in the accurate prediction of novel disease targets with biological significance. This achievement can be mainly attributed to our pioneering approach, which integrates literature evidence with heterogeneous biological networks to build a probabilistic knowledge graph. We believe that Progeni may provide a powerful and useful tool for target identification and thus facilitate the drug discovery process.

## Methods

### Data processing

We first compiled a set of heterogeneous biological networks originally collected from [17] to construct a probabilistic knowledge graph and evaluate the performance of Progeni. This set of heterogeneous biological networks contained the following information: a protein-protein interaction network downloaded from the HRPD database release 9 [40], a drug-drug interaction network and a drug-protein interaction network derived from the DrugBank database version 3.0 [41], a drug-disease association network and a protein-disease association network extracted from the Comparative Toxicogenomics Database [11], and a drug-side effect association network obtained from the SIDER database version 2 [42] (Table A in S1 Text). These networks contained a total of 12, 015 biological entities, including 1, 512 proteins, 708 drugs, 5, 603 diseases, and 4, 192 side effects. Based on these networks, a heterogeneous graph was then constructed, which contained 1, 596, 745 protein-disease associations out of 1, 895, 454 relations. In addition, we integrated two extra biological networks: a drug-drug-structure-similarity network, computed using RDKit (http://www.rdkit.org/) according to the Dice similarity of the Morgan fingerprints with radius 2 [43], and a protein-protein-sequence-similarity network, computed according to the Smith-Waterman scores [44] with the BLOSUM50 substitution matrix [45].

In our applications of predicting targets for human melanoma and colorectal cancer with experimental validation, we compiled a new dataset of heterogeneous biological networks from more recent open databases that contained larger numbers of diseases and proteins. In particular, in this new dataset, we derived a protein-disease association network from DisGe-NET [10], a protein-protein interaction network and a drug-side effect network from Decagon [46], a drug-drug interaction network and a drug-disease association network from MINER [47], and a drug-protein interaction network from both Decagon and MINER (Table B in S1 Text). Note that we used only the expert-curated data in DisGeNET, i.e., excluding those

samples inferred from indirect evidence. These networks contained a total of 27, 467 biological entities, including 9, 045 proteins, 227 drugs, 10, 111 diseases, and 8, 084 side effects. The constructed heterogeneous graph contained 77, 429 protein-disease associations out of 865, 970 relations. Note that a drug-drug-structure-similarity network and a protein-protein-sequence similarity network were also computed using the same tools as above. More details about this new network dataset can be found in S1 Text.

## The probabilistic knowledge graph

Progeni integrates literature evidence with the heterogeneous biological network data mentioned in the previous section to construct a probabilistic knowledge graph (prob-KG). The literature evidence was gathered through the curation of about three million papers from the PubMed database maintained by the United States National Library of Medicine (NLM) [48]. These papers contain abundant evidence about the associations and interactions between biological entities.

Our probabilistic knowledge graph (prob-KG) $G$ consists of nodes (representing biological entities) $v \in V$ of entity types $t \in T$ and edges (representing relations) $e \in E$ of relation types $r \in R$ defined between two entity types (e.g., proteins and diseases). Here, the entity type set $T$ is defined as $T$ = {*drug*, *target (protein)*, *side effect*, *disease*}, and the relation type set $R$ is defined as.e.g. $R$ = {*drug-drug interaction*, *drug-side effect association*, *drug-protein interaction*, *drug-disease association*, *protein-disease association*, *protein-protein interaction*, *drug-drug structure similarity*, *protein-protein sequence similarity*}.

If the relation $r \in R_s$ = {*drug-drug structure similarity*, *protein-protein sequence similarity*}, edges are formed between all pairs of entities of the corresponding types, forming a fully-connected subgraph. The edge probability $p(e)$ for an edge $e = (i, j, r)$ is then defined as the computed similarity score between entities $i$ and $j$.

If the relation $r$ is an association/interaction (e.g., *drug-protein interaction* or *protein-disease association*), an edge is formed between entities $i$ and $j$ in $G$ if an association/interaction between them is observed in the corresponding biological network. For each edge $e = (i, j, r)$, we define an edge probability $p(e)$ based on (i) the relation type $r$, (ii) the existence of the corresponding association/interaction in the biological networks, and (iii) the co-occurrence frequency between entities $i$ and $j$ in the literature evidence:

$$p(e) = (\sigma(C_r[i,j] + \alpha) - 1) \cdot \mathbb{1}_{R_c}(r) + 1, \tag{1}$$

where $\sigma(x) = \frac{1}{1+e^{-x}}$ stands for the sigmoid function, $C_r[i, j]$ represents the co-occurrence frequency between entities $i$ and $j$ in the literature evidence, $\alpha$ stands for a hyperparameter, $R_c$ = {*drug-protein interaction*, *drug-disease association*, *protein-disease association*}, and $\mathbb{1}_{R_c}(r)$ denotes the following indicator function:

$$\mathbb{1}_{R_c}(r) = \begin{cases} 1, & \text{if } r \in R_c, \\ 0, & \text{if } r \notin R_c. \end{cases} \tag{2}$$

Here, the $C_r$ values are computed via substring matching using the Trie hashing algorithm (S1 Text).

## Inference of new targets

Progeni learns the latent feature embeddings of nodes in the prob-KG and then uses them to reconstruct the knowledge graph, upon which the high-probability edges are retrieved and analyzed, thus revealing the potential target candidates. Progeni first uses the relation type-

specific graph neural networks to aggregate the neighborhood information for every node in the prob-KG and then projects the aggregated information onto a feature vector space to obtain the node embeddings. More specifically, for a relation type $r \in R$ defined between entities of types $a, b \in T$, we define a probabilistic adjacency matrix $P_r$, whose entry $P_r[i, j] = p(e)$ if a corresponding relation between entities $i$ and $j$ is observed in the biological networks and $P_r[i, j] = 0$ otherwise. Then, for all nodes of entity type $a$, a graph neural network is employed to calculate the aggregated neighborhood information matrix $Y_a$ as follows:

$$A_r = \mathbb{1}_{R_s}(r) * P_r + (1 - \mathbb{1}_{R_s}(r)) * I(P_r > 0),$$

$$\hat{A}_r = \frac{A_r}{\|A_r\|_2},$$

$$Y_a = \sum_{r \in R^a} \hat{A}_r X_b W_r,$$

(3)

where $R_s = \{$*drug-drug structure similarity*, *protein-protein sequence similarity*$\}$ stands for the set of the two similarity relation types, $\mathbb{1}_{R_s}(r)$ stands for the indicator function as defined in Eq 2, $I(P_r > 0)$ stands for a binarization of $P_r$, where positive entries of $P_r$ are mapped to one and all of its zero entries are retained, $R^a$ denotes the set of relation types defined between $a$ and any entity type (including $a$ itself), $X_b$ stands for a learnable initial embedding matrix for all entities of type $b$ connected to entities of type $a$ via edges of relation type $r$, and $W_r$ represents a learnable weight matrix defined specifically for relation type $r$.

The aggregated information matrix $Y_a$ is then combined with the initial embedding matrix $X_a$ to generate a new node embedding matrix $Z_a$ for entity type $a$ in a projection step as follows:

$$\tilde{Z}_a = ReLU((Y_a|X_a)W^1),$$

$$Z_a = \frac{\tilde{Z}_a}{\|\tilde{Z}_a\|_2},$$

(4)

where '|' stands for the concatenation operation, $ReLU(x) = \max\{x, 0\}$ represents the ReLU activation function, and $W^1$ denotes a learnable weight matrix applied to all entity types.

Progeni then reconstructs the edge probability matrix $P_r$ using the new node embedding matrices $Z_a$ and $Z_b$. Minimizing a weighted sum of the differences between the ground truth and the reconstructed matrices $P_r$s for all relation types is the optimization objective of Progeni:

$$\min_{\Theta} \sum_r \|(Z_a G_r H_r^T Z_b^T - P_r) * M_r\|_2^2,$$

$$\Theta = \{X_a | a \in T\} \cup \{W_r, G_r, H_r | r \in R\} \cup \{W^1\},$$

(5)

$$M_r[i, j] = \frac{\sigma(C_r[i, j] + \beta) - 1}{\sigma(C_r[i, j] + \alpha)} P_r[i, j] \cdot \mathbb{1}_{R_c}(r) + 1,$$

where $Z_a$ and $Z_b$ represent the node embedding matrices as defined in Eq 4, $G_r$, $H_r$ stand for the relation-type-specific projection matrices that map the node feature space to the edge probability space, $\Theta$ denotes the set of all learnable parameters, including the initial node embedding matrices $X_a$ for all entity types $a \in T$ (Eq 3), the weight matrix $W_r$ (Eq 3) and projection matrices $G_r, H_r$ for all relation types $r \in R$, and the global weight matrix $W^1$ applied to all entity types (Eq 4), and the matrix $M_r$ weights the differences between the reconstructed

and ground truth matrices of $P_r$. In the calculation of the entry $M_r[i, j]$, $\beta$ stands for a hyperparameter, and $\alpha$, $\sigma(\cdot)$, $R_c$, and $\mathbb{1}_{R_c}(r)$ are all the same as defined in Eq 1.

In the implementation of Progeni, the $\ell_2$-regularization terms on all the parameters in $\Theta$ are also added to the optimization objective to prevent overfitting the training set. Moreover, if $r \in \{$drug-drug structure similarity, protein-protein sequence similarity, drug-drug interaction, protein-protein interaction$\}$, where the corresponding matrices $P_r$ are symmetric, the constraint $G_r = H_r$ is also imposed to enforce such a symmetry.

It is also worth noting that there may still exist evidence in the literature supporting those unobserved interactions/associations in the biological networks. Nevertheless, such corresponding literature evidence is not integrated into Progeni (see Eqs 3, 4 and 5), thus preventing the potential data leakage problem during our cross-validation process.

## Cross-validation settings

We developed two cross-validation tests to evaluate Progeni and the baseline models, i.e., the entry-wise cross-validation and the cluster-wise cross-validation. In the entry-wise cross-validation, we conducted a randomly stratified split on individual entries of the edge probability matrix. In particular, the entries were divided into five folds, where the positive-to-negative ratio of each fold was roughly the same as that of the whole dataset. For each of the five iterations, we sequentially chose one fold as the test set and sampled 10% of the remaining four folds as the validation set for hyperparameter tuning (i.e., the remaining 90% constituted the training set).

In the cluster-wise cross-validation, we first conducted agglomerative clustering on the columns of the ground truth TDA matrix (i.e., the target association profiles of the diseases) according to their pairwise Jaccard similarities. We then partitioned the resulting clusters of columns. The positive-to-negative ratios of the training, validation, and test sets and their respective sizes were about the same as those in the entry-wise cross-validation scheme.

## Ablation studies

To show that integrating literature evidence into the knowledge graph was necessary for achieving better model performance, we also developed four models as control and then tested them in our ablation studies. They differ from Progeni mainly in defining the edge probability $p(e)$ and the weight $M_r[i, j]$. We first define a randomized edge probability $p'(e)$:

$$p'(e) = (u_1(\sigma(\alpha), 1) - 1) \cdot \mathbb{1}_{R_c}(r) + 1, \tag{6}$$

where $u_1(a, b)$ randomly samples a value offline from a uniform distribution between the interval $[a, b)$, and $\alpha$, $\sigma(\cdot)$, $R_c$, and $\mathbb{1}_{R_c}(r)$ are all the same as defined in Eq 1. Here, when $r \in R_c$, $p'(e)$ corresponds to a uniform sample between the interval $[\sigma(\alpha), 1)$, which is the range of all possible $p(e)$ values when $C_r[i, j]$ varies between $[0, +\infty)$ (see Eq 3).

We also define a randomized probabilistic adjacency matrix $P'_r$ in a similar manner to that of $P_r$: $P'_r[i, j] = p'(e)$ if a corresponding relation between entities $i$ and $j$ is observed in the biological networks and $P'_r[i, j] = 0$ otherwise.

We then define a randomized weight matrix $M'_r$:

$$M'_r[i, j] = \frac{u_2(\sigma(\beta), 1) - 1}{u_1(\sigma(\alpha), 1)} P'_r[i, j] \cdot \mathbb{1}_{R_c}(r) + 1, \tag{7}$$

where $\beta$ is the same as defined in Eq 5, $u_1(a, b)$ is the same as defined in Eq 6, $u_2(a, b)$ randomly samples a value offline from a uniform distribution between the interval $[a, b)$ (sampled

independently of $u_1$), and $\sigma(\cdot)$, $R_c$, and $\mathbb{1}_{R_c}(r)$ are all the same as defined in Eq 1. Here, when $r \in R_c$ and an interaction/association is observed between entities $i$ and $j$, $M_r'[i, j]$ corresponds to a uniform sample between the interval $[\sigma(\beta), 1)$, which is the range of all possible $M_r[i, j]$ values when $C_r[i, j]$ varies between $[0, +\infty)$ (see Eq 5). Also, when no interaction/association is observed between entities $i$ and $j$, $M'[i, j] = M[i, j] = 1$.

The following models were then developed as control to nullify the literature evidence integrated in the probabilistic knowledge graph: Progeni_og (i.e., no literature evidence integrated), Progeni_rp (i.e., using random edge probabilities), Progeni_rw (i.e., using random loss weights), and Progeni_rpw (i.e., using both random edge probabilities and random loss weights). We define the optimization objectives of these four models as follows:

$$\min_{\Theta} \sum_r \|Z_a G_r H_r^T Z_b^T - A_r\|_2^2, \quad \text{(Progeni\_og)} \tag{8}$$

$$\min_{\Theta} \sum_r \|(Z_a G_r H_r^T Z_b^T - P_r') * M_r\|_2^2, \quad \text{(Progeni\_rp)} \tag{9}$$

$$\min_{\Theta} \sum_r \|(Z_a G_r H_r^T Z_b^T - P_r) * M_r'\|_2^2, \quad \text{(Progeni\_rw)} \tag{10}$$

$$\min_{\Theta} \sum_r \|(Z_a G_r H_r^T Z_b^T - P_r') * M_r'\|_2^2, \quad \text{(Progeni\_rpw)} \tag{11}$$

where $\Theta$, $G_r$, and $H_r$ are all the same as defined in Eq 5, $Z_a$ and $Z_b$ are the node embedding matrices as defined in Eq 4, $A_r$ is the same as defined in Eq 3, and $P_r$, $M_r$ stand for the edge probability and weight matrices used in the Progeni model, respectively.

## Cell culture

MC38, B16F10, and CT26 cells were purchased from the American Type Culture Collection and cultured in cell incubators at 37°C with 5% $CO_2$. The cells were then cultured in the Dulbecco's modified Eagle's medium (DMEM) supplemented with 10% fetal bovine serum, 1% non-essential amino acid, 1% 4-(2-hydroxyethyl)-1-piperazineethanesulfonic acid (HEPES), 100 U/ml penicillin, and 100 μg/ml streptomycin.

## Reagents

Tanespimycin (CAS: 75747–14-7), and SNX-5422 (CAS: 908115–27-5), and Phentolamine (CAS: 65–28-1) were purchased from Abmole BioScience, Shanghai, China. All compounds were dissolved in DMSO as 10 μM or 100 μM stocks, stored at −20°C, and diluted by DMEM complete medium before use.

## Generation of the shRNA knockdown cell lines

Short hairpin RNA (shRNA)-carrying constructs were derived from the MISSION LentiPlex Mouse shRNA Library (Sigma) and then used for the knockdowns of target genes in cell lines.

Briefly, 293T cells (purchased from the American Type Culture Collection) were transfected with the transfer plasmid, psPAX2, and pMD2.G. After 48 hours, supernatants containing lentivirus were collected, filtered through a 0.45 μm membrane, and stored at −80°C before use. Next, MC38 or B16F10 cells were transduced with lentivirus in the presence of 8 μg/ml polybrene. Finally, two days after transduction, the cells were selected with puromycin until resistant clones emerged.

Targeting sequences of shRNAs used in this study are listed below.

shAdcy5:
CCGGGCTGCAGATATTCCGCTCTAACTCGAGTTAGAGCGGAATATCTGCAGCTTTTTG

shAdra2a:
CCGGGCTCATGCTGTTCACAGTATTCTCGAGAATACTGTGAACAGCATGAGCTTTTT

shEef2:
CCGGCGTGCCATCATGGACAAGAAACTCGAGTTTCTTGTCCATGATGGCACGTTTTTG

shHsp90ab1:
CCGGGCTGAACAAGACAAAGCCTATCTCGAGATAGGCTTTGTCTTGTTCAGCTTTTT

shRps6Kb1:
CCGGGCATGGAACATTGTGAGAAATCTCGAGATTTCTCACAATGTTCCATGCTTTTT

shMme:
CCGGGCAACCTATGATGATGGCATTCTCGAGAATGCCATCATCATAGGTTGCTTTTTG

Control shRNA (Non-targeting shRNA):
CCGGCAACAAGATGAAGAGCACCAACTCGAGTTGGTGCTCTTCATCTTGTTGTTTTT

## CCK-8 assay

To examine the proliferation of knockdown cell lines, we seeded equal numbers of cells in the DMEM complete medium for 36 hrs. Then, the CCK-8 assay was performed according to the manufacturer's instructions, and the optical density (OD) values were normalized to control cells. Finally, to evaluate the potential cytotoxicity, we treated the cells with inhibitors for 48 hrs, followed by a CCK-8 assay.

## Patient survival analyses

The Kaplan-Meier (KM) plots were generated by UCSC Xena (https://xenabrowser.net). The COAD and SKCM datasets from The Cancer Genome Atlas (TCGA) were selected to validate the target candidates predicted by Progeni for colorectal cancer and melanoma, respectively.

## Statistics in experimental validation

The data of the CCK-8 assay were represented as mean ± SD and compared using an unpaired Student's $t$-test. The survival curves were compared using a log-rank (Mantel-Cox) test.

## Supporting information

**S1 Text. Supplementary information file.** Figure A, Tables A-N, and other supplementary information are included. Table A: Summary of the biological networks used in our performance evaluation process and their respective data sources. Table B: Summary of the new biological networks used in the applications of Progeni to predict the target candidates of human melanoma and colorectal cancer and their respective data sources. Table C: The supplementary results on the ablation studies in the cluster-wise cross validation test (mean ± standard deviation). The results where Progeni outperformed all control methods are presented in bold.

Table D: The top 10 unobserved targets candidates predicted by Progeni for bronchitis. Table E: The top 10 unobserved targets candidates predicted by Progeni for pulmonary arterial hypertension (PAH). Table F: The top 10 unobserved targets candidates predicted by Progeni for liver neoplasms. Table G: Literature counts for the top 15 novel targets predicted by Progeni for melanoma. Table H: Literature counts for the top 15 novel targets predicted by Progeni for colorectal cancer. Table I: OD values for the B16F10 cells (melanoma) following shRNA knockdown. The results correspond to Fig 5A in the manuscript. Table J: OD values for the MC38 cells (CRC) following shRNA knockdown. The results correspond to Fig 6A in the manuscript. Table K: OD values for the B16F10 cells (melanoma) after 48h treatment with the HSP90AB1 inhibitor tanespimycin. The results correspond to Fig 5E in the manuscript. Table L: OD values for the B16F10 cells (melanoma) after 48h treatment with the HSP90AB1 inhibitor SNX-5422. The results correspond to Fig 5F in the manuscript. Table M: OD values for the MC38 cells (CRC) after 48h treatment with the ADRA2A inhibitor phentolamine. The results correspond to Fig 6E in the manuscript. Table N: OD values for the CT26 cells (CRC) after 48h treatment with the ADRA2A inhibitor phentolamine. The results correspond to Fig 6F in the manuscript. Fig A: **(A)**-**(B)**, Survival curves of the metastatic melanoma patients from The Cancer Genome Atlas (TCGA) with high or low expression of genes *HSP90AB1* (n = 50, **(A)**) and *MME* (n = 50, **(B)**), respectively. **(C)**, Survival curves of the primary melanoma patients from TCGA with high or low expression of *RPS6KB1* (n = 172).
(PDF)

## Acknowledgments

The authors thank Dr. Fangping Wan for helpful discussions. The results of this work were in part based upon data generated by the TCGA Research Network: https://www.cancer.gov/tcga.

## Author Contributions

**Conceptualization:** Chang Liu, Cuinan Yu, Dan Zhao, Fengfeng Zhou, Haidong Tang, Jianyang Zeng.

**Data curation:** Chang Liu, Cuinan Yu.

**Formal analysis:** Chang Liu, Kaimin Xiao, Yipin Lei, Tingzhong Tian.

**Funding acquisition:** Dan Zhao, Fengfeng Zhou, Haidong Tang, Jianyang Zeng.

**Investigation:** Chang Liu, Kaimin Xiao, Kangbo Lyu.

**Methodology:** Chang Liu, Kaimin Xiao, Kangbo Lyu.

**Project administration:** Jianyang Zeng.

**Resources:** Dan Zhao, Haidong Tang, Jianyang Zeng.

**Software:** Chang Liu, Kangbo Lyu.

**Supervision:** Dan Zhao, Fengfeng Zhou, Haidong Tang, Jianyang Zeng.

**Validation:** Kaimin Xiao.

**Visualization:** Chang Liu, Kaimin Xiao.

**Writing – original draft:** Chang Liu, Kaimin Xiao.

**Writing – review & editing:** Chang Liu, Kaimin Xiao, Dan Zhao, Haidong Tang, Jianyang Zeng.

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
