## [Decision Letter · Decision Letter 0]

16 Oct 2023

Dear Prof. Zeng,

Thank you very much for submitting your manuscript "A probabilistic knowledge graph for target identification" for consideration at PLOS Computational Biology.

As with all papers reviewed by the journal, your manuscript was reviewed by members of the editorial board and by several independent reviewers. In light of the reviews (below this email), we would like to invite the resubmission of a significantly-revised version that takes into account the reviewers' comments.

The reviewers appreciate the focus of the manuscript and commend the authors on providing experimental validation of the results. However, they raise several points that should be addressed to strengthen the paper, related to the data used, confirming results of lower-ranked proteins, and improving the discussion.

We cannot make any decision about publication until we have seen the revised manuscript and your response to the reviewers' comments. Your revised manuscript is also likely to be sent to reviewers for further evaluation.

Sincerely,

Stacey D. Finley, Ph.D.

Section Editor

PLOS Computational Biology

Reviewer's Responses to Questions

**Comments to the Authors:**

Reviewer #1: The review has been uploaded as an attachment.

Reviewer #2: The study introduces Progeni, a machine learning-based framework that integrates heterogeneous biological networks with literature evidence to identify effective drug targets. Progeni's robust predictive power, resilience to exposure bias, and ability to identify target candidates with biological significance in melanoma and colorectal cancer are confirmed through extensive tests and wet-lab experiments, suggesting its potential for improving drug discovery processes. The design of the validation process is sturdy; however, there are numerous major and minor issues that need to be addressed to persuade the reader.

Major issues:

1. Is there a solid rationale for constructing biological networks using outdated resources? The authors utilized DrugBank v3.0, released in 2011, while the most recent version is 5.1.10, published in 2023. Similarly, they used SIDER v3.0 from 2012 instead of the latest version 4.1, released in 2015. With the significant advancements in these fields over the past decade, the constructed networks and prediction results could vary drastically. It is strongly recommended to use the most recent version.

2. The Comparative Toxicogenomics Database contains curated and inferred drug-disease relationships. It's important for the authors to outline which type was used in their study. For accuracy, it is suggested to limit the usage to curated relationships.

3. The authors need to provide a rationale for choosing mouse cell lines over human cell lines for experimental validation, as results from the mouse model may not accurately replicate human outcomes. Furthermore, did the authors limit their calculation of protein-protein co-occurrence frequency to human proteins in the literature? Were orthologs mapped to human proteins? Was mouse protein-protein similarity also considered? To mitigate these potential complications, it is strongly advised to restrict the model to human only.

4. Co-occurrence might not be the most reliable metric for substantiating protein-disease associations, as the correlation could be opposing, i.e., inhibitory or stimulatory. How are these diverse relation types managed within the probabilistic knowledge graph?

5. Only the top-ranked proteins were considered for experimental validation; however, lower-ranked proteins might also be effective in treating melanoma or colorectal cancer. Therefore, it would be advantageous to conduct experiments on some lower-ranked proteins and known treatment targets for melanoma or colorectal cancer as well.

Minor issues:

1. Initially, I found the term "target" perplexing as it typically refers to a drug target in drug discovery studies. The term "biomarker" might be more intuitive, or alternatively, it may be clearer to refer to it as a "disease target" or "treatment target".

2. Which substitution matrix was applied when assessing protein-protein sequence similarity?

3. The arrangement of subfigures (the sequence of a, b, c, and d) in Figure 4 differs from the layout observed in Figures 2 and 3.

4. How is the high or low expression of genes defined within TCGA?

5. For consistency with Figure 5, it is suggested to present the survival curves for primary colorectal cancer patients from TCGA with high or low expression of ADRA2A and EEF2 in Figure 6, irrespective of the results showing significant differences or not.

**Have the authors made all data and (if applicable) computational code underlying the findings in their manuscript fully available?**

Reviewer #1: Yes

Reviewer #2: Yes

PLOS authors have the option to publish the peer review history of their article (what does this mean?). If published, this will include your full peer review and any attached files.

Reviewer #1: No

Reviewer #2: **Yes: **Liang-Chin Huang
---

## [Decision Letter · Decision Letter 1]

30 Nov 2023

Dear Prof. Zeng,

Thank you very much for submitting your manuscript "A probabilistic knowledge graph for target identification" for consideration at PLOS Computational Biology. As with all papers reviewed by the journal, your manuscript was reviewed by members of the editorial board and by several independent reviewers. The reviewers appreciated the attention to an important topic. Based on the reviews, we are likely to accept this manuscript for publication, providing that you modify the manuscript according to the review recommendations.

While many reviewer comments have been addressed, some issues remain. The authors are advised to edit the manuscript text to address the remaining concerns, which are centered around validation of the findings. Additional analyses can be included. Additionally, the Discussion can be revised to better reflect the conclusions that can be drawn given experimental validation. Furthermore, there are minor comments that should be addressed.

Sincerely,

Stacey D. Finley, Ph.D.

Section Editor

PLOS Computational Biology

Stacey Finley

Section Editor

PLOS Computational Biology

Reviewer's Responses to Questions

**Comments to the Authors:**

Reviewer #1: I have no major comments left, as they all have been adequately addressed. The discussion is mostly what I was looking for, and some of the overfitting issues have been made as small as possible. I enjoy the extra addition to the Github as well.

Reviewer #2: While the authors have addressed the majority of the comments and suggestions put forth by the reviewers, a few questions remain unresolved or unattended to in the Discussion section.

Major issues:

1. While the authors have updated their datasets, furnished new target candidates, and presented them in the supplementary tables, further validation is lacking. I appreciate the fact that experimental validation may require an extended period of time, but I would encourage the authors to analyze the expressions of these candidates in relation to survival probability at the very least. Furthermore, utilizing the newly added relations in the updated datasets as a test set for the existing model could demonstrate the robustness of the methodology employed.

2. I concur that utilizing a mouse model is sufficient for the experimental validation of the predicted targets. Nevertheless, there are still certain issues that either remain unresolved or have not been addressed: when determining the frequency of protein-protein co-occurrence and their similarity, were human or mouse proteins employed in the calculations? Were orthologs, not just those of the mouse, but others as well, mapped to human proteins?

3. I again recognize that experimental validation could be a lengthy process, however, certain methods such as survival analyses are necessary to justify the relevance of low-ranked proteins. The authors have asserted that "low-ranking proteins will have near-0 correlations, and thus have low relevance to the disease". Without proper validation, this assertion could be questionable since these low-ranking proteins might indeed have significant relevance to the diseases.

4. None of the experimental evidence (such as the images of clones) has been attached.

Minor issue:

The figures and tables are missing titles and captions.

**Have the authors made all data and (if applicable) computational code underlying the findings in their manuscript fully available?**

Reviewer #1: Yes

Reviewer #2: Yes

PLOS authors have the option to publish the peer review history of their article (what does this mean?). If published, this will include your full peer review and any attached files.

Reviewer #1: **Yes: **Brandon J. Bongers

Reviewer #2: **Yes: **Liang-Chin Huang

Figure Files:

Data Requirements:

Reproducibility:

References:

---

## [Decision Letter · Decision Letter 2]

21 Jan 2024

Dear Prof. Zeng,

Thank you very much for submitting your manuscript "A probabilistic knowledge graph for target identification" for consideration at PLOS Computational Biology. As with all papers reviewed by the journal, your manuscript was reviewed by members of the editorial board and by several independent reviewers. The reviewers appreciated the attention to an important topic. Based on the reviews, we are likely to accept this manuscript for publication, providing that you modify the manuscript according to the review recommendations.

The reviewers appreciate the revised manuscript. The remaining outstanding action item is to provide the experimental images from which the optical density values were taken. These can be added to the supplementary material and will enhance the rigor and reproducibility of the work.

Sincerely,

Stacey D. Finley, Ph.D.

Section Editor

PLOS Computational Biology

Stacey Finley

Section Editor

PLOS Computational Biology

Reviewer's Responses to Questions

**Comments to the Authors:**

Reviewer #2: The authors have solved all the issues and provided supporting evidences.

**Have the authors made all data and (if applicable) computational code underlying the findings in their manuscript fully available?**

Reviewer #2: Yes

PLOS authors have the option to publish the peer review history of their article (what does this mean?). If published, this will include your full peer review and any attached files.

Reviewer #2: **Yes: **Liang-Chin Huang

Figure Files:

Data Requirements:

Reproducibility:

References:

---

## [Editor Report · Decision Letter 3]

24 Feb 2024

Dear Prof. Zeng,

We are pleased to inform you that your manuscript 'A probabilistic knowledge graph for target identification' has been provisionally accepted for publication in PLOS Computational Biology.

Best regards,

Stacey D. Finley, Ph.D.

Section Editor

PLOS Computational Biology

Stacey Finley

Section Editor

PLOS Computational Biology

---

## [Editor Report · Acceptance letter]

28 Mar 2024

PCOMPBIOL-D-23-00810R3 

A probabilistic knowledge graph for target identification

Dear Dr Zeng,

I am pleased to inform you that your manuscript has been formally accepted for publication in PLOS Computational Biology. Your manuscript is now with our production department and you will be notified of the publication date in due course.

With kind regards,

Zsuzsanna Gémesi
